# Anti-elite attitudes and support for independent candidates

Pablo Argote[1]☯, Giancarlo Visconti[2]☯ *

**1** Department of Political Science and International Relations, University of Southern California, Los Angeles, California, United States of America, **2** Department of Political Science, Pennsylvania State University, University Park, Pennsylvania, United States of America

☯ These authors contributed equally to this work.
* gvisconti@psu.edu

**Data Availability Statement:** Replication files: https://dataverse.harvard.edu/dataset.xhtml?persistentId=doi:10.7910/DVN/XGHULL.

**Funding:** Project Funded by the College of Liberal Arts at Purdue University. The funders had no role in study design, data collection and analysis,

## Abstract

The ideological dispute between left and right has dominated the political discussion for decades in multiple countries across the globe. However, in recent years people vs. elite debates have replaced traditional ideological conflicts in explaining voters' electoral decisions. In this paper, we investigate whether anti-elite attitudes contribute to a key political outcome: the increase in successful independent candidates. We implement a conjoint experiment in Chile, where anti-elite sentiments and the number of successful independent politicians are currently prominent. We find that preferences for independent candidates largely increase among voters with anti-elite orientations. In a context where traditional parties face difficult times, such beliefs are key to understanding the factors behind support for candidates with no party affiliation. This insight is important because electing independent politicians can promote a personalistic style of politics, undermine democratic accountability, and foster conflict between the executive and legislative branches.

## Introduction

Anti-elite attitudes—particularly those promoting a moral conflict between "the people" and "the corrupt elite" [1, 2]—have become more prominent around the world over the last decade. Such attitudes are now prevalent and politically relevant in places as diverse as Spain, Turkey, the United States, and Venezuela [3–6]. While the battle of left vs. right has long dominated the political conversation, the conflict between the people and elites has become a key factor in explaining voters' electoral choices in recent years [7].

Another relevant political phenomenon is the electoral success of independent candidates (i.e., individuals who run for office without the support of, or an affiliation with, a political party) in a variety of political settings [8–10]. For example, in 2022, Rodolfo Hernandez ran as an independent in Colombia and almost won the election in the second round, obtaining 47% of the votes. In 2016, Patrice Talon, an independent businessman, was elected president of Benin by defeating the incumbent prime minister. In 2018, the independent Salome Zourabichvili was elected the first woman president of Georgia. This political transformation is happening in a context of a decline in party identification [11], erosion of party brands [12], reduction of party system institutionalization [13], and an increase in alienation from the

decision to publish, or preparation of the manuscript.

**Competing interests:** The authors have declared that no competing interests exist.

party system [14]. In Appendix A in S1 File, we expand our definition of independents and differentiate them from outsiders and populist candidates [15].

Previous studies have demonstrated that anti-elite attitudes correlate with feelings of dissatisfaction [16] or discontent with the democratic system [17]. Since independent candidates typically antagonize established political elites, voters might consider them an attractive alternative for expressing their desire for change in the political system. Indeed, independent candidates might be perceived as more apt to represent the will of the people rather than partisan interests.

In this paper, we posit that there is an underexplored connection between anti-elite beliefs and support for independent candidates. Previous research has studied these variables in isolation. The studies on the *consequences of anti-elite attitudes* have mainly focused on how these orientations induce people to vote for anti-elite parties [18] or to support anti-elite candidates who belong to traditional parties [7]. When examining *why people vote for independent candidates*, the research has mainly focused on institutional or socioeconomic variables. For example, they show how various electoral rules affect independent candidates' performance [15, 19] and how adverse conditions (e.g., economic contractions, inflation, or even disasters) can shift support from traditional parties to independent politicians [20, 21].

Latin America has faced these two types of political transformations in recent years. First, anti-elite attitudes have become quite prevalent among citizens [22]. A common chant in social protests is: *Que se vayan todos* (they all must go), which symbolizes people's negative attitudes toward traditional political elites and a yearning for a different type of politician. Second, a boom in the number of independent candidates has occurred in the context of a large crisis of representation that has alienated citizens [23], increased social protests [24], in light of political parties with weak roots in society [25]. The average number of votes received by independent candidates in first-round presidential elections in the region has increased from 2% before 2000 to 15% after 2015 (using data from [26]). A country that has experienced both phenomena is Chile. In the last decade, anti-elite rhetoric and antipathy toward the political system have become common [27], and the number of independent candidates and elected officials have dramatically increased.

Our research design is based on an original survey of 3,965 adults living in Chile, implemented before the 2021 presidential election. To capture the outcome of interest (i.e., voting for independents), we used a pre-registered conjoint experiment to randomize the political affiliation of several hypothetical presidential candidates. Before exposing respondents to the conjoint, we measured the prevalence of anti-elite attitudes. Then, we compared anti-elite and non-anti-elite respondents regarding their preferences for independent candidates. We also asked survey respondents to describe partisan and independent politicians in their own words, which allows us to know what voters understand about affiliated and unaffiliated candidates. We discuss the pre-analysis plan in Appendix B in S1 File.

One of the main methodological challenges associated with investigating the link between anti-elite beliefs and support for independent candidates is that people with anti-elite attitudes might be different from those without such views. To address this concern, we use inverse probability weighting (IPW) to generate groups of anti-elite and non-anti-elite respondents with similar distributions of multiple observed variables that previous studies have identified as key predictors of voters' electoral choices in Chile [28–30] (see Appendix G in S1 File).

We first found a strong preference for independent candidates over members of a political coalition. Respondents with prior anti-elite beliefs largely drive this difference: they are 8 percentage points more likely to vote for independents than those without such beliefs; these results remain unaltered when using IPW. These findings help us better understand key elements of contemporary political discussion. For instance, they offer a sound explanation of

why independent candidates can become more attractive to certain types of voters in countries where the anti-elite rhetoric has become part of the everyday political discussion.

## Context: Protests and elections in Chile

In October 2019, a wave of student protests motivated by increased subway fares led to widespread street violence, vandalism, and police repression. Over the next few days, hundreds of thousands of people took to the streets to protest and demand deep and systemic changes to address problems related to pensions, health, education, and inequality. In response, Congress agreed to initiate a process to change the Pinochet-era constitution.

The 2019 social outburst has been interpreted as an anti-elite manifestation largely explained by protesters' frustration and anger at a system that was perceived to benefit only a small subset of the Chilean population [31, 32]. Party identification has experienced a drastic decline in Chile in recent years [33] and citizens show a high animosity toward traditional parties [34]. In this context, political parties have struggled to connect with voters, a process that started before 2019 [35]. The protests that year represented a total divorce between the political establishment and the people. Consequently, most ordinary citizens believe parties do not understand their needs and are thus unable to represent their interests [36].

In 2021, Chile held its first presidential election after the protests. The three candidates who received the most votes in the first round were: the far-right José Antonio Kast, who ran on the backlash effects of the unrest; the left-wing Gabriel Boric, who led the student protests of 2011–2012; and Franco Parisi, a former economics professor, and businessman who first ran in 2013 as an independent candidate with a strong anti-elite campaign.

The newly formed ad hoc "People's Party" (Partido de la Gente) backed Parisi's 2021 campaign, highlighting that he was not part of the political establishment and denounced traditional parties. This anti-elite appeal resonated with an important segment of the electorate: he obtained more than 10% of the votes in both presidential elections he contested. The emergence of candidates such as Franco Parisi illustrates how anti-elite sentiments can help independent politicians gain support. This political phenomenon is not unique to Chile. For example, Peru and Colombia have also experienced massive social protests recently and have had independent candidates with notable electoral results, such as Pedro Pablo Kuczynski (in 2011) and Rodolfo Hernandez (in 2022), respectively.

## Data, measures and experimental design

We commissioned Netquest, a market research firm with ample experience in Latin America, to administer the survey to members of its online panel. All participants provided informed consent before participating in the survey. No deception was used, and participation in the survey was confidential. The Institutional Review Boards of Purdue and Columbia University reviewed and approved all procedures. We implemented this pre-registered online survey between October 29 and November 20, 2021. To generate a sample that resembles the Chilean electorate, we adopted quotas for age, gender, socioeconomic status, and region. In Appendix K in S1 File, we compare the distribution of the demographic variables in our sample to those of the 2017 census for which comparable data is available.

### Measuring anti-elite attitudes

To capture anti-elite beliefs, we rely on recent evidence that voters view politics as a conflict between both "left" and "right" and between "the people" and "the elite" [7]. We asked respondents about both dimensions simultaneously to acknowledge that these are the key dimension of politics in multiple countries, including Chile [37]. The question reads as follows:

Below you will read two statements; which one is closer to your ideas?

1. The main division in society is between the people and the elite.

2. The main division in society is between the left and the right.

We asked subjects to select between these two options to provide a clear context for having anti-elite attitudes, since it is easy for respondents to report having such beliefs in isolation (i.e., when they are not compared to other dimensions to organize politics, such as the left–right spectrum). We used the responses to this question to generate a binary indicator of anti-elite attitudes (1 = the main division in society is between the people and the elite, and 0 otherwise). The statement about the main division in society as being between "the people" and "the elite" comes from the Agnes Akkerman index [38].

As a robustness check, we used an alternative measure of anti-elite attitudes using the question, "below, you will read two statements; which one is closer to your ideas? i) legislators should follow the will of the people when making laws, or ii) legislators should follow their own knowledge and opinions when making laws." Respondents who selected the first alternative were considered to have anti-elite attitudes on the basis that supporters of anti-elite ideas prefer using what the people want (rather than technical expertise) to guide policy-making. Finally, we construct a third indicator that combines reporting anti-elite attitudes on both the first and second measures. Our conclusions are not conditional on how we capture anti-elite orientations (see Appendix E in S1 File). Importantly, all these measures of anti-elite attitudes were obtained before any experimental analysis was conducted, so they are pre-treatment measures.

## Conjoint experiment

In the survey, we included a conjoint experiment that presented profiles of two hypothetical presidential candidates. For each candidate, we simultaneously randomized four attributes: i) political coalition member or independent, ii) occupation (lawyer, teacher, or street vendor), iii) age (35, 45, 55, or 65 years old), and iv) gender (man or woman).

Our main theoretical interest relies on the first attribute, which allows us to compare coalitions with different ideological positions to independent politicians. This attribute has five categories—Chile's four political coalitions: Chile Podemos Más (UDI, RN, Evopoli, etc.), Nuevo Pacto Social (DC, PS, PPD, etc.), Apruebo Dignidad (Frente Amplio, PC, etc.), Frente Social Cristiano (Partido Republicano, Partido Conservador Cristiano), and independents (no party affiliation). We assigned a higher weight to the independent category: respondents were exposed 50% of the time to an independent and 50% to a member of a political coalition.

We included the other attributes to generate realistic profiles and had no theoretical expectations for their effects. We repeated the experiment five times for each respondent to increase statistical power. For the analysis, we clustered standard errors at the respondent level. Table 1 displays an example of two profiles.

**Table 1. Example of two profiles.**

| Attribute | Candidate 1 | Candidate 2 |
|---|---|---|
| Coalition | Chile Podemos Más (UDI, RN, Evopoli, etc.) | Independent (No party or coalition) |
| Occupation | Lawyer | School teacher |
| Age | 45 | 35 |
| Gender | Man | Woman |

Since this study was conducted during the electoral campaign but before the presidential election, to mitigate the risk that respondents could get confused when choosing between hypothetical candidates during an electoral period, in an additional check, we interacted the conjoint attributes with the number of days left until election day and found no effect. These results, reported in Appendix J in S1 File, demonstrate that the campaign did not affect how people chose between hypothetical candidates in our study. Moreover, it is important to discuss upfront whether people's choices in this experimental setting actually reflect their actual ones. On the one hand, we acknowledge a limitation in generalizing our findings to every electoral choice, as people are evaluating hypothetical candidates. However, previous research has compared the findings from conjoint experiments with behavioral benchmarks, showing that both analyses generated similar results in countries such as Switzerland [39] and Chile [21].

## Empirical strategy

Our quantity of interest is the average marginal component effect (AMCE) [40]—the traditional estimand of conjoint experiments—which corresponds to the effect of being independent vs. non-independent, averaged over the joint distribution of the remaining attributes. We use the following basic regression equation to capture respondents' preference for independents:

$$Y_i = \beta_0 + \beta Independent_i + \sum_{j=1}^{2} \delta_j Occupation(j)_i + \sum_{j=1}^{3} \tau_j Age(j)_i + \eta Gender_i + \epsilon \qquad (1)$$

Where $Y_i$ represents a binary choice for respondent $i$. The coefficients $\beta$ correspond to the marginal effect of having an independent candidate vs. one who belongs to a political coalition. Our second regression model interacts with the anti-elite indicator with all the attributes:

$$Y_i = \beta_0 + \beta Independent_i + \lambda(Anti-elite)_i + \gamma(Independent * (Anti-elite))_i + \qquad (2)$$

$$\sum_{j=1}^{2} \delta_j Occupation(j) + \sum_{j=1}^{2} \theta_j (Occupation(j) * (Anti-elite))_i + \sum_{j=1}^{3} \tau_j Age(j) + \qquad (3)$$

$$\sum_{j=1}^{3} \phi_j (Age(j) * (Anti-elite))_i + \eta Gender_i + \alpha(Gender * (Anti-elite))_i + \epsilon \qquad (4)$$

Where $\beta$ indicates how being an independent candidate affects non-anti-elite respondents' vote choices. $\beta + \gamma$ refers to the same effect but for anti-elite respondents. In Appendix I in S1 File, we conduct multiple diagnostic checks for the conjoint analysis.

## Results

The left side of Fig 1 displays a coefficient plot for the attributes of interest (i.e., independent vs. partisan candidate) using the entire sample. Respondents are 12.3 percentage points more likely to choose an independent candidate than a coalition member, a significant and substantive effect (95% CI: [11.4, 13.4]). The right side of Fig 1 compares the treatment effect of being an independent candidate for anti-elite and non-anti-elite survey participants. 54% of the sample is considered anti-elite, and 46% non-anti-elite (using the first measurement approach previously described). Participants with anti-elite beliefs exhibit a clear preference for independent candidates: they are 16.2 percentage points (95% CI: [14.5, 17.5]) more likely to vote for an independent than for a partisan candidate. By contrast, respondents without such

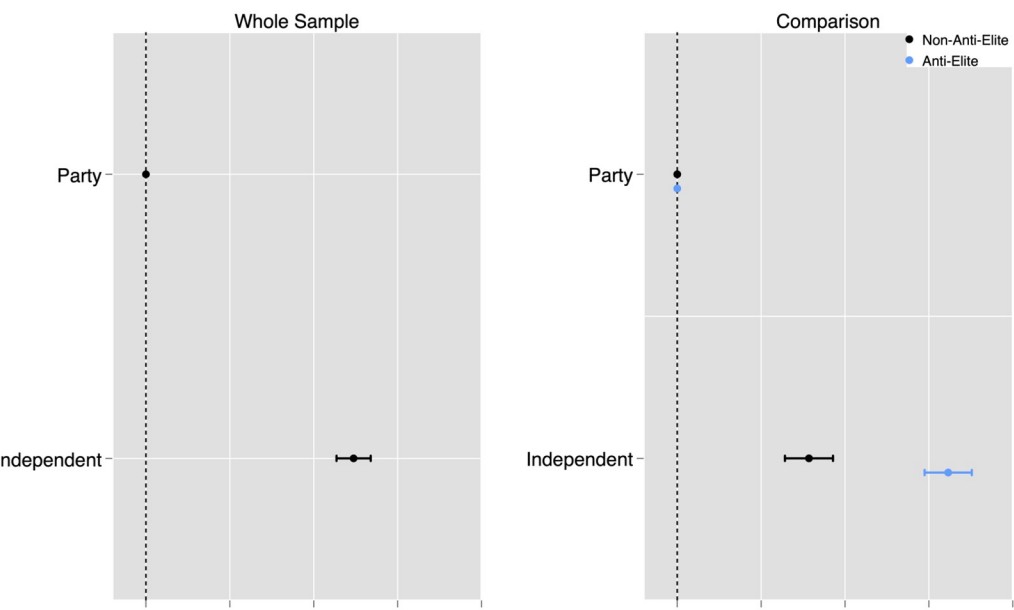

**Fig 1. Preference for independents by anti-elite beliefs.** The outcome is the preference for a given candidate. The other conjoint attributes are omitted (see Appendix C in S1 File for the complete interacted and non-interacted results). The dots represent the point estimates, and the lines 95% confidence intervals. Standard errors are clustered at the respondent level. Number of observations: 39,650 (3,965 survey participants).

beliefs have only a moderate preference for those candidates: they are 7.8 percentage points (95% CI: [6.4, 9,2]) more likely to vote for an independent than for a partisan candidate. Finally, the difference between the two groups is substantive and statistically significant. The interaction term shows that anti-elite respondents are 8.3 percentage points more likely to choose an independent candidate than non-anti-elite ones (95% CI: [6.3, 10.3]). We provide all the results in table format and the full conjoint experiment in Appendix C in S1 File.

## Robustness checks

To address the concern that comparing anti-elite and non-anti-elite respondents can mask differences between the two groups that could be driving our results, we use inverse probability weighting (IPW) in Appendix G in S1 File. First, we provide evidence of a covariate balance between anti-elite and non-anti-elite respondents on key observed characteristics such as region, education, gender, age, ideology, and previous turnout after using IPW. Second, we show that our main conclusions do not change after making these adjustments to improve the between-group comparison. As mentioned above, we also use alternative measures of anti-elite to rule out the possibility that our findings can be explained by how we capture anti-elite attitudes. Moreover, we added control variables to the main specifications and used marginal means instead of AMCE. Our conclusions are the same when implementing these extra analyses (see Appendices F, E, and H in S1 File).

In Appendix K in S1 File, we address potential external validity problems since our sample under-represents less educated Chileans. We weighted the observations using census counts based on the combination of region, education, age, and gender. We created cell weights (adjusted according to the census counts of the described combination) and rake weights (based on the separate marginal distributions of each variable). We found that the results are somewhat amplified, as the preference for independent candidates rises to 17 percentage

points [95% CI: 15.6, 20.5], whereas the interaction coefficient—the difference between anti-elite and non-anti-elite—reaches 10 percentage points [95% CI: 7.4, 14.3]. Giving more weight to less educated respondents likely increased support for independent candidates.

## Text analysis

In the previous section, we documented a strong preference for independent candidates, mostly driven by respondents with anti-elite attitudes. However, it is also important to document the words associated with "Independents" and "Party Members" to see what comes into people's minds when facing these cues. Table 2 displays the most frequently mentioned words translated into English.

Their answers indicated clear negative attitudes toward elites; they used words such as "thieves" and "corrupt." They characterized independents using descriptions such as "hope" or "change," suggesting that such candidates symbolize values opposite to the political establishment (the word "none" refers to people describing independents as politicians with no partisan affiliation). In Appendix L in S1 File, we provide word clouds in Spanish. These results complement our main findings by showing how different people's views are when evaluating partisan and independent candidates. As a result, anti-elite citizens will have incentives to prefer candidates associated with new ideas (i.e., independents) rather than with the traditional establishment (i.e., party-members).

## Discussion and conclusion

We explore how anti-elite attitudes affect support for independent candidates. Since we identified a general preference for independent over partisan candidates, other factors may also explain why people support unaffiliated politicians, such as the overall crisis of representation in the country [41]. However, survey respondents with anti-elite attitudes were more than twice as likely to support independent candidates when compared with individuals without anti-elite orientations. This finding is robust to using inverse probability weighting to generate covariate balance between anti-elite and non-anti-elite voters, which illustrates the relevance of anti-elite sentiments in explaining support for independent politicians.

The rise of anti-elite attitudes not only promotes the emergence of anti-elite parties, such as Podemos in Spain, or candidates with anti-elite rhetoric embedded into traditional parties, such as Donald Trump in the US. Such beliefs can also mobilize votes for independent politicians such as Chile's Franco Parisi and Colombia's Rodolfo Hernandez.

**Table 2. Most repeated words: Independents and party members.**

| Party Members | Independents |
|---|---|
| Thiefs | None |
| Corrupt | Ideas |
| Same | People |
| Group | New |
| Corruption | Alone |
| Always | Support |
| People | Change |
| Group | Hope |
| Interests | Opportunistic |
| Sell-out | Freedom |
| Lawyers | Exist |

Electing independent candidates is not innocuous and can bring diverse impacts to the political system. On the more negative side of consequences, they might promote a personalistic style; undermine democratic norms and institutions [42]; foster conflicts between the executive and legislative branches due to the absence of a parliamentary party [43]; and jeopardize the quality of representation, accountability, and responsiveness [19]. On the more positive side, unaffiliated politicians may also incorporate new approaches and discussions into the political arena [44]. Consequently, it becomes very important to understand why people support this type of political leader.

Although institutional and socioeconomic variables are key for understanding the emergence of independent candidates, they have a limited ability to explain their recent success in national elections. Consequently, it is crucial to study the attitudes underlying the preferences for these candidates. In addition, future research could explore other sentiments and orientations contributing to the rise of independents, which can help us better understand why these types of politicians are more likely to emerge in certain places than others.

## Supporting information

**S1 File. Supporting information.**
(PDF)

## Acknowledgments

Authors are listed alphabetically. We thank Ryan Carlin, Francisca Castro, John Marshall, and Daniel Tavana for their helpful comments and suggestions. We preregistered this study at Open Science Framework before we finished collecting our data. All errors are our own.

## Author Contributions

**Conceptualization:** Pablo Argote, Giancarlo Visconti.

**Data curation:** Pablo Argote, Giancarlo Visconti.

**Formal analysis:** Pablo Argote, Giancarlo Visconti.

**Funding acquisition:** Pablo Argote, Giancarlo Visconti.

**Investigation:** Pablo Argote, Giancarlo Visconti.

**Methodology:** Pablo Argote, Giancarlo Visconti.

**Project administration:** Pablo Argote, Giancarlo Visconti.

**Resources:** Pablo Argote, Giancarlo Visconti.

**Software:** Pablo Argote, Giancarlo Visconti.

**Supervision:** Pablo Argote, Giancarlo Visconti.

**Validation:** Pablo Argote, Giancarlo Visconti.

**Visualization:** Pablo Argote, Giancarlo Visconti.

**Writing – original draft:** Pablo Argote, Giancarlo Visconti.

**Writing – review & editing:** Pablo Argote, Giancarlo Visconti.

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
