## [Decision Letter · Decision Letter 0]

22 Jun 2023

PONE-D-23-15838Anti-Elite Attitudes and Support for Independent CandidatesPLOS ONE

Dear Dr. Visconti,

Thank you for submitting your manuscript to PLOS ONE. After careful consideration, we feel that it has merit but does not fully meet PLOS ONE’s publication criteria as it currently stands. Therefore, we invite you to submit a revised version of the manuscript that addresses the points raised during the review process.

We look forward to receiving your revised manuscript.

Kind regards,

Pablo Henríquez, Ph.D.

Academic Editor

PLOS ONE

Journal Requirements:

"Project Funded by the College of Liberal Arts at Purdue University. "

6. Please amend your list of authors on the manuscript to ensure that each author is linked to an affiliation. Authors’ affiliations should reflect the institution where the work was done (if authors moved subsequently, you can also list the new affiliation stating “current affiliation:….” as necessary).

Reviewers' comments:

Reviewer's Responses to Questions

**Comments to the Author**

1. Is the manuscript technically sound, and do the data support the conclusions?

Reviewer #1: Yes

Reviewer #2: Yes

2. Has the statistical analysis been performed appropriately and rigorously? 

Reviewer #1: Yes

Reviewer #2: Yes

3. Have the authors made all data underlying the findings in their manuscript fully available?

Reviewer #1: Yes

Reviewer #2: Yes

4. Is the manuscript presented in an intelligible fashion and written in standard English?

Reviewer #1: Yes

Reviewer #2: Yes

5. Review Comments to the Author

Reviewer #1: I read the paper with great interest. I support the paper moving toward publication.

1. I think the connection between the election independent candidates and democratic accountability, responsiveness, and executive relations need to be better explained. There is nothing inherently bad about independent politicians, and there is little reason why their election should be a major normative concern. I think maybe the issue is that some independent candidates are anti-system candidates, and that is what the analysis is picking up on.

2. The words "anti-establishment" and "anti-elite" are used interchangeably. Authors should pick their terms, define them, and stick with them.

3. I like the items for capturing anti elite attitudes.

4. More should be done with the text analysis in the main body of the paper.

Reviewer #2: I would like to congratulate the authors on this well-written manuscript that investigates the effect of anti-establishment attitudes on the support for independent candidates in Chile using a conjoint experiment. The objective of the study is straightforward, and the manuscript is concise and focused. Overall, I enjoyed reading this paper and I recommend its publication. However, I would like to raise some points that should be addressed.

Firstly, I suggest being more cautious about whether the measure captures anti-establishment “attitudes”. The measure relies on a trade-off question that may be easily influenced by what is currently on top of respondents’ mind. They are asked to indicate which of the following is closer to their ideas: 1) the main division in society is between the people and the elite or 2) the main division in society is between the left and the right. If anti-establishment attitudes are more salient, they will pick the former, if left-right conflict is more salient, then they will pick the latter even if their actual attitudes do not change over time. So, it seems like this measure may capture something quite volatile rather than relatively stable anti-establishment attitudes. Authors check whether days to election influences the support for independent candidates. However, I think it would be beneficial to check whether anti-establishment attitudes vary over time and whether, once days to election accounted for, the support for independent candidates among anti-elite and non-anti-elite participants is influenced. Additionally, it would be helpful to examine the correlations between all measures of anti-establishment used in the paper.

Secondly, regarding the design, I understand that the initial idea was to prime anti-establishment attitudes using three different treatments (crisis of representation, electoral malfeasance, inequality). The explanation for deviating from the from pre-analysis plan is logical, but I am a little confused about the new design. Were anti-establishment attitudes measured before or after respondents were experimentally primed? If measured before experimental priming, then respondents already thought about whether the division of left versus right or of the people versus the elite is more salient/important before experimental priming. This could also explain the null results reported in Appendix B, as the control group is also primed before answering this question. If measured after experimental priming, then it is problematic, as the main independent variable would be influenced by the design, which would probably lead to an overestimation of the observed effect. Authors should clarify these.

More than 50 percent report anti-establishment attitudes in the sample. Isn’t that a little too high? How does it compare to other measured? It would be valuable to discuss whether this may be influenced by the way it is measured and how it compares to findings in the existing literature. Providing context in this regard would be beneficial.

Lastly, I would acknowledge the limitations of external validity in the findings (hypothetical candidates). In the sample, everyone is more likely to vote for an independent candidate than for a partisan candidate (even those without anti-elite beliefs). It is worth elaborating on how the results generalize to the real context and what patterns we would expect to observe in real world? And do we see those?

I hope this is useful and wish the best.

6. PLOS authors have the option to publish the peer review history of their article (what does this mean?). If published, this will include your full peer review and any attached files.

Reviewer #1: No

Reviewer #2: No

---

## [Author Response · Author response to Decision Letter 0]

24 Jul 2023

Please see the following document: "memo_PONE.pdf."

---

## [Decision Letter · Decision Letter 1]

5 Sep 2023

PONE-D-23-15838R1Anti-Elite Attitudes and Support for Independent CandidatesPLOS ONE

Dear Dr. Visconti,

Thank you for submitting your manuscript to PLOS ONE. After careful consideration, we feel that it has merit but does not fully meet PLOS ONE’s publication criteria as it currently stands. Therefore, we invite you to submit a revised version of the manuscript that addresses the points raised during the review process.

We look forward to receiving your revised manuscript.

Kind regards,

Pablo Henríquez, Ph.D.

Academic Editor

PLOS ONE

Journal Requirements:

Reviewers' comments:

Reviewer's Responses to Questions

**Comments to the Author**

1. If the authors have adequately addressed your comments raised in a previous round of review and you feel that this manuscript is now acceptable for publication, you may indicate that here to bypass the “Comments to the Author” section, enter your conflict of interest statement in the “Confidential to Editor” section, and submit your "Accept" recommendation.

Reviewer #1: All comments have been addressed

Reviewer #2: All comments have been addressed

2. Is the manuscript technically sound, and do the data support the conclusions?

Reviewer #1: Yes

Reviewer #2: Yes

3. Has the statistical analysis been performed appropriately and rigorously? 

Reviewer #1: Yes

Reviewer #2: N/A

4. Have the authors made all data underlying the findings in their manuscript fully available?

Reviewer #1: Yes

Reviewer #2: Yes

5. Is the manuscript presented in an intelligible fashion and written in standard English?

Reviewer #1: Yes

Reviewer #2: Yes

6. Review Comments to the Author

Reviewer #1: In the abstract, the authors claim that anti-elite sentiments have increased. I don't know if there is data showing that. If they don't have it, then cut that claim from the abstract.

Final sentence of the abstract is a bit bland. Any candidate, independent or not, can have "diverse consequences" a range of things. What is it uniquely about independent candidates that is of concern or interest?

Reviewer #2: I would like to thank the authors for responding to my comments in the previous round. Authors' responses are fair and convincing. Hence, I can recommend its publication. I do however have two minor comments that authors should see to before submitting their final manuscript.

1) The discussion on external validity is essential but comes out of nowhere. I think it will be better as a footnote.

2) The text analysis part is interesting but somehow disconnected. As it is now, authors just describe the table without linking it to what precedes it. Can the authors say something in how this complements the main findings and what to take away from these most used words in a sentence or two?

Thanks for the opportunity to review this interesting piece and congrats to the authors.

7. PLOS authors have the option to publish the peer review history of their article (what does this mean?). If published, this will include your full peer review and any attached files.

Reviewer #1: No

Reviewer #2: No

---

## [Editor Report · Decision Letter 2]

12 Sep 2023

Anti-Elite Attitudes and Support for Independent Candidates

PONE-D-23-15838R2

Dear Dr. Visconti,

We’re pleased to inform you that your manuscript has been judged scientifically suitable for publication and will be formally accepted for publication once it meets all outstanding technical requirements.

Kind regards,

Pablo Henríquez, Ph.D.

Academic Editor

PLOS ONE
---

## [Editor Report · Acceptance letter]

4 Oct 2023

PONE-D-23-15838R2 

Anti-Elite Attitudes and Support for Independent Candidates 

Dear Dr. Visconti:

I'm pleased to inform you that your manuscript has been deemed suitable for publication in PLOS ONE. Congratulations! Your manuscript is now with our production department. 

Kind regards, 

on behalf of

Dr. Pablo Henríquez 

Academic Editor

PLOS ONE